# Characterization, Stress Response and Functional Analyses of Giant River Prawn (*Macrobrachium rosenbergii*) Glucose-Regulated Protein 78 (Mr-grp78) under Temperature Stress and during *Aeromonas hydrophila* Infection

**DOI:** 10.3390/ani11103004

**Published:** 2021-10-19

**Authors:** Prapansak Srisapoome, Tanya Ju-Ngam, Ratree Wongpanya

**Affiliations:** 1Laboratory of Aquatic Animal Health Management, Department of Aquaculture, Faculty of Fisheries, Kasetsart University, Chatuchak, Bangkok 10900, Thailand; radin_the@hotmail.com; 2Center of Advanced Studies for Agriculture and Food, Kasetsart University Institute for Advanced Studies, Kasetsart University, Bangkok 10900, Thailand; 3Center of Excellence in Aquatic Animal Health Management, Faculty of Fisheries, Kasetsart University, Chatuchak, Bangkok 10900, Thailand; 4Department of Biochemistry, Faculty of Science, Kasetsart University, Bangkok 10900, Thailand; fscirtw@ku.ac.th

**Keywords:** *Aeromonas hydrophila*, giant river prawn, gene knockdown, *grp78*, heat/cold temperature shock, heat-shock protein

## Abstract

**Simple Summary:**

Glucose-regulated protein 78 (*grp78*) is classified as a member of the Hsp70 subfamily. This protein functions as a key factor in signal transduction associated with the unfolded protein response (UPR) in the endoplasmic reticulum (ER) during cellular stress and protects against cell damage caused by toxic chemicals, oxidative stress, Ca^2+^ depletion, programmed cell death and various infectious conditions. To investigate this crucial mechanism in giant river prawn (*Macrobrachium rosenbergii*), we analyzed the biological function of prawn *grp78* at the molecular level in this study. The regulation of this gene was intensively analyzed under normal bacterial infection and heat/cold-shock inductions. A functional analysis of this gene under heat and infectious stress conditions was performed by gene knockdown. The information obtained in the current study clearly indicates the crucially significant roles of *grp78* in the cellular stress responses of the target experimental animal under various stress conditions.

**Abstract:**

The endoplasmic reticulum (ER) is an organelle important for several functions of cellular physiology. This study identified the giant river prawn’s *glucose-regulated protein 78* (*Mr-grp78*), which is important for ER stress mechanisms. Nucleotide and amino acid analyses of *Mr-grp78*, as compared with other species, revealed the highest similarity scores with the *grp78* genes of crustaceans. An expression analysis by quantitative RT-PCR indicated that *Mr-grp78* was expressed in all tissues and presented its highest expression in the ovary (57.64 ± 2.39-fold), followed by the gills (42.25 ± 1.12), hindgut (37.15 ± 2.47), thoracic ganglia (28.55 ± 2.45) and hemocytes (28.45 ± 2.26). Expression analysis of *Mr-grp78* mRNA levels under *Aeromonas hydrophila* induction and heat/cold-shock exposure was conducted in the gills, hepatopancreas and hemocytes. The expression levels of *Mr-grp78* in these tissues were highly upregulated 12 h after bacterial infection. In contrast, under heat- and cold-shock conditions, the expression of *Mr-grp78* was significantly suppressed in the gills at 24–96 h and in the hepatopancreas at 12 h (*p* < 0.05). A functional analysis via *Mr-grp78* gene knockdown showed that *Mr-grp78* transcription in the gills, hepatopancreas and muscle strongly decreased from 6 to 96 h. Furthermore, the silencing of this gene effectively increased the sensitivity of the tested prawns to heat- and pathogenic-bacterium-induced stress. The results of this study clearly demonstrate the significant functional roles of *Mr-grp78* in response to both temperature and pathogen treatments.

## 1. Introduction

Heat-shock proteins (Hsps) are a group of proteins with essential roles in dealing with stresses based on their function as molecular chaperones, which are crucial for repairing damaged proteins to restore their normal properties via protein refolding [1]. In general, Hsps are scientifically classified according to their important structures, molecular weights and functions into numerous protein families, including several smaller Hsps, Hsp40, Hsp60, Hsp70, Hsp90 and Hsp100 [2]. Relatively low levels of Hsp expression are generally observed during the normal activities of living cells, such as growth, development and apoptosis [3]. These molecules can be activated by several stress conditions, such as abnormal physical factors, pathogen infections and critical environmental stimuli [4,5].

In cellular biology, the endoplasmic reticulum (ER) is a critical organelle in all eukaryotic cells and functions to control protein translation and transport, Ca^2+^ level homeostasis and protection against programmed cell death [6,7]. Under environmental conditions, the occurrence of ER stress resulting from imbalanced Ca^2+^ homoeostasis and unfolded protein accumulation in the lumen stimulates signal transduction cascades collectively referred to as the unfolded protein response (UPR) [8,9]. Three key UPR signal activator proteins have been identified: PKR-like ER kinase (PERK), activating transcription factor 6α/β (ATF6) and inositol requiring enzyme 1α/β (IRE1). These components initiate three different response mechanisms, and all of these mechanism branches aim to mitigate the load of mis- or unfolded proteins and to completely drive homeostasis of ER protein [9,10]. These responses are crucially initiated by glucose-regulated protein 78 (grp78) (heat-shock 70 kDa protein 5 (HSPa5) or immunoglobulin heavy-chain-binding protein (BiP)), which belongs to the Hsp70 subfamily found in the lumen of the ER, where it functions in the responses to toxic chemicals, oxidative stress and Ca^2+^ depletion, and this protein thereby further induces cellular stress in the ER lumen [11,12]. Additionally, this protein protects cells against cell death caused by abnormal ER homoeostasis conditions, which may decrease oxyradical accumulation and stabilize mitochondrial function, similar to the functions of an anti-apoptotic protein [13]. Many studies have reported that grp78 can play a role in the immune system by controlling immunoglobulin heavy chain protein and α2-macroglobulin signal propagation to defend the host against foreign pathogens [14,15,16].

Giant river prawn (*Macrobrachium rosenbergii*) is known as an important crustacean for economic trade found in many countries around the world, particularly in Central and Southeast Asia, and exhibits acceptable cultivation characteristics, such as an appealing taste, high market demand, high resistance to stress, and good adaptability to estuarine and freshwater environments [17]. The giant river prawn has been classified as a eurythermal animal that can survive under broad-range temperature fluctuations in the range of 14 to 35 °C [18]. However, giant river prawn aquaculture has recently faced various problems, particularly related to rapid changes in culture conditions, which generally cause giant river prawns to be suppressed and easily infected with several pathogens [19,20,21]. Among these pathogens, *Aeromonas hydrophila* is responsible for severe problems in aquaculture conditions because it can cause serious diseases in various economically important species, including giant river prawn. This bacterium can cause opportunistic disease and induce clinical signs when its hosts are under stress conditions or are co-infected with various pathogens [22]. It was also shown that *A. hydrophila*–injected prawns exhibited several changes in histopathology in different target organs, and these alterations included hyperplasia, focal necrosis, muscular damage, a mild or massive hemocyte reaction and hemolytic infiltration [23].

Very few studies have provided detailed descriptions of crustacean *grp78* genes; in fact, information is only available for Chinese white shrimp (*Fenneropenaeus chinensis*) [24], black tiger shrimp (*Penaeus monodon*) [25] and Pacific white shrimp (*Litopenaeus vannamei*) [26]. To further elucidate the biological functions of this protein in crustaceans, cDNA encoding a *grp78* gene from giant river prawn was cloned and characterized in the current study, and its expression in response to *A. hydrophila* challenge and heat/cold stress conditions was intensively investigated. A functional analysis of this gene was performed by using a gene-knockdown approach. The results of the current study provide important information on the cellular stress response and immune defense mechanisms that could be further applied to improve giant river prawn aquaculture.

## 2. Materials and Methods

### 2.1. Animals

Healthy two hundred giant river prawns (47.25 ± 5.54 g) were obtained from a grow-out prawn farm located in Nakhon Pathom Province, Thailand. Two days before transportation, the animals were randomly diagnosed for external parasites and bacterial infection, using routine laboratory protocols at the farm. The acclimatization of all tested animals was conducted for 7 days in a 1000 L fiberglass tank under clean, fully aerated conditions. The water temperature was maintained at 30 ± 2 °C. All experimental prawns were fed commercial pellet feed (36% crude proteins, 4% fat, 8% carbohydrate and 12% moisture) 3 times a day, at a level equal to 3–5% of their body weight. Prior to scarifying for any purposes, the experimental animals were always anesthetized with ice for exactly 1 min.

### 2.2. PCR with 5′ Rapid Amplification cDNA Ends (5′RACE) 

Total RNA and mRNA were isolated from the gills of a healthy giant river prawn, using TRIzol reagent (Gibco, Waltham, MA, USA) and a QuickPrep Micro mRNA Purification Kit (Amersham Biosciences, Piscataway, NJ, USA), respectively, following the manufacturer’s protocols. First-strand cDNA for 5’ RACE was then synthesized, using the prepared mRNA. The mRNA (1 µg/µL) was reverse-transcribed into 5’ ready-to-use RACE 1st-strand DNA by a BD Smart^TM^ RACE cDNA Amplification Kit (BD Biosciences, San Jose, CA, USA; Clontech, San Francisco, CA, USA). Briefly, 1 µL of mRNA was mixed with 1 µL of 5’-CDS primer, 1 µL of BD SMART II A oligo and 2 µL of distilled water. This mixture was incubated at 70 °C for 2 min and cooled at 0 °C for 2 min. One microliter of DTT, 2 µL of 5X first-strand buffer, 1 µL of dNTP mix and 1 µL of BD Power Script Reverse Transcriptase were incubated at 42 °C for 90 min. This 1st-strand cDNA was used as a target to amplify the nucleotide sequences in the 5’ direction of the cDNA. The specific primer R_GFP_grp78 (Table 1) was designed based on a partial cDNA sequence (BG2420, EL609443) obtained from an expressed sequence tag cDNA library of giant river prawn because this sequence is homologous to the *grp78* genes of other animal species. Each round of RACE PCR with each primer pair (final volume of 25 µL) was conducted strictly according to the 5’ RACE PCR instructions and touchdown PCR specified by the manufacturer’s protocol to recover the full-length cDNA of this gene. The first five cycles consisted of 94 °C for 30 s and 72 °C for 3 min, and the second set of 5 cycles involved amplification at 94, 70 and 72 °C for 30 s, 30 s and 3 min, respectively. Subsequently, 27 cycles of 94, 68 and 72 °C for 30 s, 30 s and 3 min were performed to obtain the target PCR fragments.

### 2.3. Cloning and Characterization of the Full-Length cDNA Encoding the grp78 Gene of Giant River Prawn

The obtained PCR products as described in Section 2.2 were isolated by a QIAquick^®^ gel extraction kit (QIAgen^®^, Germantown, MD, USA), and all cDNA cloning steps and sequencing were carefully conducted according to a previous study [27]. All obtained sequences were analyzed for homology with sequence information available in the GenBank database (http://www.ncbi.nlm.nih.gov) (accessed on 9 January 2021) via the BLASTX and BLASTN programs. The 5’ RACE and partial sequences from the counterpart clone were aligned to obtain the full-length cDNA encoding the *grp78* gene of giant river prawn.

The full-length cDNA encoding the *grp78* gene was again searched for homology by the above methods. The 5’ and 3’ untranslated regions (UTRs) and open reading frames (ORFs) of the obtained cDNA were analyzed by using ORF Finder (Open Reading Frame Finder, http://www.ncbi.nl.nih.gov/gorf/gorf.html) (accessed on 9 January 2021). The DAS transmembrane prediction program (http://www.sbc.su.se/~miklos/DAS) (accessed on 9 January 2021) was further used to analyze the presence of a leader peptide. All amino acid motifs were predicted by the Simple Modular Architecture Research Tool program (http://smart.cmbl-heidelberg.de/) (accessed on 19 January 2021). The Compute p*I*/Mw tool of ExPASy (http://www.expasy.org/tools/pi_tools.html) (accessed on 19 January 2021) was employed to analyze the theoretical isoelectric point (p*I*) and molecular weight of the predicted proteins.

To analyze the multiple sequence alignments of the target known *grp78* genes, deduced amino acid sequences of other known *grp78* genes obtained from various animal species in the GenBank database were retrieved and carefully conducted, using CLUSTAL W (http://ebi.ac.uk/Tools/clustalw/index.html) (accessed on 29 April 2021) [28]. Both nucleotide and amino acid sequences were neatly compared, using MatGAT (Matrix Global Alignment Tool) version 2.02 (http://bitincka.com/ledion/matgat/) (accessed on 29 April 2021).

### 2.4. Phylogenetic Analysis

Phylogenetic tree analysis of the metazoan *grp78* genes was carried out by using the amino acid sequence of the giant river prawn *grp78* gene and other known *grp78* sequences from various vertebrates, invertebrates, plants, fungi and protozoa (see phylogenetic tree). The DAS transmembrane prediction server was used to remove the leader sequences, all grp78 sequences were aligned with the same methods as described above [28], and the phylogenetic tree was created by using MEGA version 6 with 1000 bootstrap replicates [29]. Human heat-shock protein 90α2 was employed as an outgroup.

### 2.5. Determination of the Distribution of grp Genes in Various Tissues of Healthy Giant River Prawns Based on Quantitative Real-Time RT-PCR (qRT-PCR)

One healthy female and one healthy male maintained as described in Section 2.1 were used. Total RNA was extracted from 13 different tissues, namely the gills, eyestalk, heart, foregut, hepatopancreas, subcuticular epithelium, midgut, hindgut, muscle, and thoracic ganglion of both sexes, the testis and vas deferens of the male prawn, and the ovary of the female prawn, using previously described methods [27]. Subsequently, 0.5 micrograms of total RNA from each tissue from each sex was mixed with the qualified testis, vas deferens and ovary RNA. A RevertAid First Strand cDNA Synthesis Kit (Fermentas, Waltham, MA, USA) was used to synthesize the first-strand cDNAs from the total RNA from each tissue separately. The construction methods and conditions strictly followed the manufacturer’s instruction manual.

The expression levels of the giant river prawn *grp78* gene in 13 different tissues were analyzed by qRT-PCR, using 1 µL of 1st-strand cDNA of each tissue and Brilliant III SYBR Green qPCR Master Mix (Stratagene, La Jolla, CA, USA) in an Mx3005P real-time PCR system equipped with version 4.0 of the corresponding analytical software according to the manufacturer’s recommended protocol (Stratagene, La Jolla, CA, USA). PCR was prepared and contained 1 µL of first-strand cDNA, 10 µL of 2× SYBR Green qPCR Master Mix, 5 µL of distilled water and 1 µL of each specific primer pair (including RTF_GFP_grp78 primer and RTR_GFP_grp78) (Table 1) to make a final volume of 18 µL. The *grp78* gene expression levels in each tissue were normalized relative to the expression of the *β-actin* gene, using the specific primers shown in Table 1. PCR analysis followed the same methods described by Reference [27]. The relative copy numbers of the target mRNA of each gene were calculated according to the 2^−ΔCt^ method [30].

### 2.6. Analysis of the grp78 mRNA Expression Response in Giant River Prawns under Aeromonas hydrophila Stimulation

A virulent strain of *A. hydrophila* (AQAH0101) isolated from infected prawns was inoculated into 15 mL of tryptic soy broth (TSB, Merck, Kenilworth, NJ, USA) and then incubated at 30 °C in an incubation shaker for 24 h. Centrifugation at 800× *g* for 15 min was performed to collect bacterial pellets and washed with sterile 0.85% NaCl twice. The concentration of *A. hydrophila* was dissolved in 0.85% NaCl to an optical density at 540 nm of 0.1 to obtain the original stock at 1 × 10^9^ CFU/mL. The bacterial solution was further prepared with 0.85% NaCl to obtain a final concentration of 1 × 10^7^ CFU/mL.

In this experiment, 3 groups of thirty healthy prawns in Section 2.1 were separately selected and maintained in 3 different 1000 L fiberglass tanks for 7 days as described above. Each prawn in the first and second experimental groups was intramuscularly injected with 0.1 mL of 2 different concentrations at 1 × 10^7^ or 1 × 10^9^ CFU/mL of the previously prepared *A. hydrophila*, respectively, in the muscle area between the 3rd and 4th pleura. Every prawn in the third tank was injected with 0.1 mL of 0.85% NaCl, using a similar injection used to treat the prawn in the other 2 groups. All prawns in each group were kept and cared for in their acclimatization containers with the same preparation as previously described, and their behaviors and morbidity were noted every day. After 0, 3, 6, 12, 24, 48 and 96 h, 3 prawns from each group were randomly selected for harvesting of their gills, hepatopancreas and hemocytes, using the same protocol described above. First-strand cDNA of all extracted total RNA at different time courses was synthesized by using the same methods in the previous section. The first-strand cDNA from each tissue collected from the different groups at different time points was subjected to qRT-PCR, using methods similar to those described in Section 2.5.

The relative copy numbers of *grp78* transcripts in each tissue of all injection groups of giant river prawns at different time courses were determined, and the expression level determined at an initial time in each tissue was used as a calibrator. The relative *grp78* gene expression was statistically determined by using one-way analysis of variance (ANOVA) and Duncan’s new multiple range test (DMRT). Significant difference was considered when *p* < 0.05.

### 2.7. Response Analyses of grp78 mRNA in Giant River Prawns to Heat- and Cold-Shock Induction

Ninety of the prawns described in Section 2.1 were chosen and maintained under the same above-described conditions for 7 days. The water temperature in each experimental tank was controlled at low fluctuation levels of 30.0 ± 0.4 °C, using an automatic heater controller (HOPARTM K-339) (Chosion, Taicang, Jiangsu, China). After 7 days of acclimatization, 8 prawns each in separate groups were transferred to 3 different containers. In containers 1 and 2, the water temperature was set to 30.0 ± 0.4 and 35.0 ± 0.7 °C, respectively, and the other conditions were maintained as in the acclimatization step. In the other tank, the prawns were raised in a cold room, and the water temperature was set to 25.0 ± 0.4 °C. The behaviors and mortality of the prawns were recorded daily for 96 h.

After 0, 3, 6, 12, 24, 48 and 96 g, 3 of the 24 prawns in each tank were killed for the isolation of total RNA from their gills, hepatopancreas and hemocytes, and first-strand cDNA synthesis was then conducted by using the same above-described methods. qRT-PCR analysis of the *grp78* gene in the organs of the prawns belonging to each temperature group collected at different time points was conducted by using the same steps as in the previous analyses. Subsequently, the relative expression ratios of the *grp78* gene were statistically tested by using an analytical method similar to that described in Section 2.6.

### 2.8. Gene Knockdown Analysis of the Mr-grp78 Gene

In this experiment, the gene-specific primers Mr-HSP78T7_F and Mr-HSP78T7_R (Table 1) were created to target a nucleotide sequence template of a T7 promotor and duplicate the target cDNA obtained as described in Section 2.2, using the same protocol described above. All PCR products were further cloned and sequenced with the same methods described above. The particular double-stranded RNAs of *Mr-grp78* (ds*Mr-grp78*) and the reference control gene (green fluorescent protein or ds*GFP*) were produced and purified, using a T7 RiboMAX™ Express RNAi System (Promega Corporation, Madison, WI, USA).

To set up the conditions for gene knockdown analysis, one thousand premature giant river prawns (3.14 ± 0.42 g) were raised under the same conditions, using the previously mentioned practices. Randomly selected animals (20 prawns) were transferred and acclimatized in three 10 L glass tanks for 7 days. Every prawn prepared in the three 10 L glass tanks was subjected to one of the following treatments (intramuscularly injected): in containers 1, 2 and 3, the target animals were individually induced by intramuscular injection with 10 µL of phosphate buffer saline (PBS, pH 7.4), PBS+3 µg ds*Mr-grp78* or PBS+3 µg dsGFP. After injection at h 0, 6, 12, 24 and 48, 3 prawns in each group were randomly selected, and the gill, hepatopancreas and muscle tissues of each prawn were used for total RNA extraction and first-strand cDNA synthesis, according to the above-described methods. Furthermore, qRT-PCR analysis of the *Mr-grp78* transcripts at different time points was performed by using the protocol described in Section 2.5.

### 2.9. Effects of Mr-grp78 Gene Silencing under High Temperature Stress and during A. hydrophila Infection

In this experiment, groups of 10 prawns, which were stocked as described in Section 2.8, were raised in 9 different 10 L glass tanks with a static water temperature of 30 ± 1.2 °C for 7 days. Subsequently, each prawn in tanks 1–3 (PBS control group), 4–6 (GFP control group) and 7–9 (ds*Mr-hsp78* knockdown) was induced with intramuscular injection of 10 µL of PBS, PBS+3 µg of ds*Mr-grp78* or PBS+3 µg of ds*Mr-grp78*, respectively. The prawns were then returned to their containers, in which the temperature of water was accurately maintained at 35.0 ± 0.7 °C, and all other conditions were maintained as described in Section 2.7. The behaviors and mortality of the prawns in each treatment group were recorded for 7 days, and significant differences in these data were tested according to the same process described in Section 2.6.

### 2.10. Effects of Mr-grp78 Gene Silencing on A. hydrophila (AH) Infection

In this experiment, prawns were maintained in 9 different 10 L glass tanks with a static water temperature of 30 ± 1.2 °C for 7 days as described in the previous section. On day 8, each prawn in tanks 1–3 (ds*GFP* control group), 4–6 (ds*GFP*+AH control group) and 7–9 (dsMr-grp78+AH group) was induced with intramuscular injection of 10 µL of PBS+3 µg of dsGFP, PBS+3 µg of dsGFP or PBS+3 µg of dsMr-grp78. The prawns were then returned to their tanks, in which the temperature of water was accurately maintained at 30 ± 1.2 °C. After 2 h, every prawn in tanks 1–3 was intramuscularly injected with 10 µL of PBS, whereas the prawns in tanks 4–6 and 7–9 were intramuscularly injected with 10 µL of PBS containing 1 × 10^7^ CFU/mL *A. hydrophila* (prepared as described in Section 2.6). After the second injection, the behaviors and mortality of the prawns in each treatment group were recorded for 7 days, and significant differences in mortality were analyzed by using the methods described in the previous section.

## 3. Results

### 3.1. Analysis of the Mr-grp78 cDNA of Giant River Prawn

In the present study, the full-length cDNA encoding giant river prawn grp78 was successfully recovered by using information from EST clones containing partial cDNAs of clone BG2420 and its 5*’*-RACE sequence fragment. The full-length cDNA of the giant-river-prawn *grp78* gene was designated “*Mr-grp78*” (GenBank accession number KM081680) (Figure 1).

Multiple alignment sequence analysis revealed that the amino acid structure of Mr-grp78 was different from that of giant river prawn hsc70 and hsp70 (AY446497 and AY466445); however, these structures comprised three signatures of the hsp70 family, including the CY sequence, ATP/GTP binding site motifs A and an ER homolog sequence. In contrast, the C-terminus of Mr-grp78 contained a KDEL motif, which was also observed in all ER proteins, whereas two other hsp70s contained MEEVD motifs (Figure 2).

### 3.2. Nucleotide and Amino Acid Sequence Analyses and Phylogenetic Tree Construction of Mr-grp78

Comparison analyses of nucleotide and amino acid identity and similarity demonstrated that *Mr-grp78* showed 67.1–95.9%, 76.4–77.6% and 68.6–77.9% similarity to the known amino acid sequences of grp78, hsp70 and hsc70, respectively, from other species. *Mr-grp78* presented notable similarity scores of 95.6% and 95.9% with *grp78* of Chinese white shrimp and Pacific white shrimp, respectively (Appendix A).

An evolutionary phylogenetic tree of Mr-grp78 with other reported species HSPs found in the GenBank database was demonstrated. The phylogenetic tree included 38 HSPs of eukaryotes and prokaryotes, and the results showed that the two major groups of the eukaryote and prokaryote hsp70 gene families were clearly clustered (Figure 3). Within the eukaryotic hsp70 family, the two subgroups were clearly clustered into the hsp70-hsc70 and grp78 groups. *Mr-grp78* was clearly grouped within the *grp78* cluster and was closely related to the *grp78* genes of Pacific white shrimp and Chinese white shrimp (Figure 3).

### 3.3. mRNA Level Expression of the grp78 Gene in Various Tissues of Giant River Prawn by qRT-PCR Analysis

The expression patterns of the *Mr-grp78* gene in various organs of healthy animals measured by qRT-PCR showed that *Mr-grp78* was detected in all tested tissues and presented a particularly high expression in the ovary (57.64 ± 2.39), gills (42.25 ± 1.12), hindgut (37.15 ± 2.47), thoracic ganglia (28.55 ± 2.45) and hemocytes (28.45 ± 2.26), whereas the other tissues showed moderate and low expression levels (Figure 4).

### 3.4. Expression Analyses of the Mr-grp78 mRNAs after A. hydrophila Induction

After injection with two concentrations of *A. hydrophila* (1 × 10^7^ and 1 × 10^9^ CFU/mL) at different time points, the *Mr-grp78* expression level in the gills was significantly upregulated at the early time points of 3 and 12 h after injection (hI) by 2.03- and 2.14-fold, respectively, in the group treated with 1 × 10^9^ CFU/mL *A. hydrophila* and was significantly upregulated at 96 hI (1.96-fold) in the group treated with 1 × 10^7^ CFU/mL (Figure 5A). In the hepatopancreas, the mRNA expression of *Mr-grp78* after 1 × 10^7^ CFU/mL *A. hydrophila* treatment was slightly significantly increased at 3 and 24 hI by 2.57- and 1.53-fold, respectively, whereas the mRNA expression of *Mr-grp78* under treatment with 1 × 10^9^ CFU/mL *A. hydrophila* showed a peak of 13.89-fold at 12 hI (Figure 5B). The *Mr-grp78* expression pattern in hemocytes was significantly decreased at the early time points of 3 (0.46-fold) and 6 hI (0.55-fold) in the group treated with 1 × 10^9^ CFU/mL *A. hydrophila*, and the expression levels were increased at 12 hI in the groups treated with both concentrations (6.29-fold and 7.80-fold; Figure 5C). During this induction, the control prawn did not demonstrate abnormal clinical signs throughout the study period, whereas the 2 doses of bacterial solution significantly induced lethargic movement and anorexia at 24–96 hI. However, no mortality was observed throughout the trials.

### 3.5. Expression Analyses of Mr-grp78 mRNA in Response to Heat and Cold Shocks

The qRT-PCR analysis revealed the expression patterns of *grp78* mRNA in the tested organs at various time points during exposure to heat/cold-shock temperature conditions (Figure 6). In the gills, the *Mr-grp78* expression pattern was slightly upregulated (2.14-fold) at 3 h after treatment (hT) at 35 °C and downregulated at 24–96 hT (Figure 6A). In the hepatopancreas, the *Mr-grp78* mRNA expression level showed a significant overall decrease at all different time points during treatment at 25 °C and was significantly increased at 24–48 hT during exposure to 35 °C (Figure 6B). The expression level of hemocyte *Mr-grp78* was detectable at the early time points of 3 and 6 hT (4.74-fold and 1.74-fold) during treatment at 35 °C, while the expression level was decreased by exposure to 25 and 35 °C and was significantly downregulated at 12 hT (Figure 6C). Additionally, no mortality was observed at any of the temperatures at 96 hT. However, at 12–96 hT, low temperature (25 °C) induced lethargic activity in the experimental prawn, whereas high temperature (35 °C) first induced vigorous swimming at 12–24 hT and then resulted in slow movement until 96 hT.

### 3.6. Silencing Analysis of Mr-grp78 mRNA under Heat Stress and A. hydrophila Infection

In this experiment, *Mr-grp78* mRNA expression in the gills, hepatopancreas and muscle was strongly silenced. Early effects were found at 6 h in the gills and hepatopancreas, and later effects were observed at 24 h in muscle. A low (4.44 ± 0.14 log copy number/100 ng) or high (7.22 ± 0.54) concentration of *Mr-grp78* was significantly knocked down from 6 or 24 h, until 96 h in the experiment (Figure 7A–C).

The effects of *Mr-grp78* gene silencing were investigated under heat shock and *A. hydrophila*–infection conditions. The results revealed that prawns subjected to *Mr-grp78* gene knockdown were more sensitive to heat (35 °C) treatment than the prawns belonging to the PBS and GFP control groups, and these three groups exhibited significantly different mortality rates of 53.33 ± 11.55, 23.33 ± 5.77 and 0.00%, respectively, at 96–168 h (Figure 8A).

Similar results were observed in the *A. hydrophila*–infection trial. In this experiment, the group of prawns treated with dsMr-grp78 and *A. hydrophila* exhibited the highest mortality of 100.0 ± 0.0% at 96–168 h, whereas the prawns injected with dsGPF alone or dsGPF and *A. hydrophila* showed lower mortality rates of 16.67 ± 11.55 and 53.33 ± 15.28%, respectively (Figure 8B).

## 4. Discussion

### 4.1. Characterization of Mr-grp78

In this research work, a novel cDNA encoding *Mr-grp78* was first characterized in giant river prawns. Structural analysis of the Mr-grp78 protein showed a leader peptide at the N-terminus that could localize to the extracellular environment. However, this peptide forms an amphiphilic helix to functionally maintain cellular systems under normal conditions [31]. A structural analysis revealed that the deduced Mr-grp78 amino acid sequence contained three conserved Hsp70 protein family amino acid signatures, a *p*-loop (ATP–GTP binding site A) and an ER homolog region [32]. ATP–GTP binding site A in the N-terminal ATPase domain is found in all Hsp70 families and plays a functional role in controlling chaperone-mediated folding [33]. However, the structure [AS]-[ED]-[AGT]-Y-[LI]-[GS]-[KQ]-[TKP] of ATP–GTP binding site A of Mr-grp78 was AEAYLGKP, which showed similarity to grp78 of Chinese white shrimp (AEAYLGKP) [23] and was closely related to two cytosolic Hsp70s of giant river prawn (AEAFLGST in Mr-HSC70 and AEAYLGKT in Mr-HSC70) [34]. In addition, Mr-grp78 showed a leader sequence at the N-terminal region and a KDEL tetrapeptide motif at the C-terminal region, which confirmed the ER-targeting strategy of this protein. The KDEL protein family, such as protein disulfide isomerase, Grp78 and calreticulin, plays crucial roles in ER functions associated with the correct and precise folding of nascent proteins in the secretory pathway [35,36].

Homological analyses indicated that *Mr-grp78* exhibited the highest identity and similarity scores to crustacean *grp* genes, supporting the existence of high gene conservation among organisms. A phylogenetic tree analysis confirmed the Hsp homology results. *Mr-grp78* was clustered within the Hsp70 family and was highly evolutionarily related to crustacean groups, similar to the findings of a previous study [37]. *Mr-grp78* is closely related to the *grp78* subfamily, particularly crustacean members, as revealed by the tree of Chinese white shrimp *grp78* [24]. Our results suggest that the Hsps from each family present highly conserved structures from invertebrates to higher vertebrates that are consistent with their various functions in cellular systems.

### 4.2. Expression Analysis of Mr-grp78

An expression analysis of the tissue distribution of the *Mr-grp78* gene indicated that *Mr-grp78* expression was ubiquitously distributed in all tested tissues. In living organisms, cellular proteins are normally synthesized to control normal functions in the body. These functions are affected by grp78 stimulation to regulate the folding of nascent proteins [38,39]. Additionally, the highest expression was observed in the ovary, which showed similar results to those of previous studies in Chinese white shrimp and black tiger shrimp [24,25], and this finding suggests that grp78 is crucial for oocyte development in crustaceans.

### 4.3. Expression Analysis of Mr-grp78 under A. hydrophila Induction

Based on currently available information, *A. hydrophila* can clearly be classified as a major causative agent of severe diseases found in the aquaculture industry in a broad range of economic organisms in an opportunistic manner [22,23]. During the infection stage, Hsps generally play important roles in the normal maintenance of cellular functions by stabilizing and refolding proteins to function in early immune responses to invading pathogens [40]. Many Hsps are involved in the implementation of these crucial mechanisms, and these Hsps include Hsp90, grp78, Hsp70, Hsp60 and Hsp27, which are activated and further moved to the extracellular environment of target cells [41,42,43,44].

In the current research, the transcriptional levels of the *Mr-grp78* gene in hemocytes were reduced at 3–6 h after *A. hydrophila* infection. grp78, with the cochaperone Hsp40, controls the removal of excessive Ca^2+^ from the ER lumen and may be disturbed by pathogenic effects [34,39,45]. In addition, grp78 plays an important role in suppressing the host cell defense mechanism by reducing oxidative stress; therefore, decreasing the grp78 level is a cellular defense mechanism for increasing the efficient activities of reactive oxygen species (ROS) [13]. However, *Mr-grp78* was highly significantly expressed in response to the two tested concentrations of *A. hydrophila* in each tested organ at 12–24 hI, particularly in hemocytes. This result indicates that defense-related organs play a key role in phagocytosis by releasing ROS to regulate pathogenic multiplication; however, this cellular mechanism is critical for increasing the signs of apoptosis in these immune cells [46]. The upregulation of Hsps, particularly grp78, safeguards surviving cells against ROS reactions [47,48]. Furthermore, grp78 can assist cells in opposing apoptosis by stabilizing mitochondrial function, which is a necessary component of the ER-induced apoptosis pathway [11,49]. Thereafter, Mr-grp78 downregulation was found in the tested tissues of *A. hydrophila*–infected prawns at 24–48 hI. The observed transcriptional expression was concisely similar to some reports discovered in crustaceans. For instance, a decrease in regulation in Chinese white shrimp was found in lymphoid organs at 24 h after virus injection [24]. This result may be correlated with the mechanism through which the pathogen load in the host body is controlled and decreased by humoral and cellular factors during the pathogenic resistance process, which induces the cells to enter a quiescent state.

### 4.4. Analysis of Mr-grp78 Expression under Heat and Cold Temperature Conditions

In aquaculture pond conditions, controlling the temperature is one of the most difficult tasks. Sudden temperature changes of 3–5 °C cause aquatic animals to become stressed and to experience shock and subsequent death [50]. Such fluctuations strongly affect the homeostasis of all aquatic animals, which are poikilothermic species [51]. The fluctuation of water temperature causes serious physiological alterations in these species and can have many consequent effects, particularly on metabolism, growth rates, the reproductive system, immune responses and sensitivity to invasion [19,20,52].

Very few studies have examined *grp78* expression in crustaceans in response to heat or cold shock. Only one study with Chinese white shrimp clearly showed significant upregulation of the *grp78* gene in shrimp within 1–4 h of exposure to a temperature of 35 °C [24]. Under higher-temperature conditions, a direct effect on aquaculture ponds has been shown to occur by decreasing the oxygen levels in water and enhancing the toxicity of some water qualities that affect the metabolism of all aquatic animals [53]. Furthermore, heat-shock conditions can effectively induce cellular stress and thereby affect the misfolding of protein synthesis in aquatic animals [54]. In a previous study of the heat-shock responses of giant river prawn Hsp70 and Hsc70, Liu et al. (2004) demonstrated that giant river prawn Hsp70 is rapidly upregulated to a highly significant level, whereas giant river prawn Hsc70 is not expressed [34]. Normally, Hsp70, Hsc70 and grp78 are classified as belonging to the Hsp70 family, but these proteins show very different expression patterns after heat-shock stress. Under unstressed conditions, the Hsp70 mRNA transcript usually exhibits lower levels than Hsc70 or grp78. Under heat shock stress, Hsp70 expression is markedly upregulated within a short time and remains increased for several hours, whereas Hsc70 shows a stable mRNA level when it is not induced, similar to *grp78*, and shows very low expression at 2 to 3 h [24,55,56].

In our study, the difference in the expression of *Mr-grp78* indicated that it plays an important role against heat and cold shock in the host. These effects agree with the findings of this study, which indicates the significant upregulation of *Mr-grp78* in gills at early periods to stabilize respiratory function and absorb increased amounts of oxygen under rapid physicochemical fluctuations in water. Previously, some research demonstrated cognate transcriptional expression in the high-temperature tolerance of invertebrate Hsp molecules, which included grp78 of the calanoid copepod *Eurytemora affinis* [57]. In the hepatopancreas, non-significant expression of *Mr-grp78* was observed during 3–6 hT. Generally, the hepatopancreas of crustaceans shows slower thermal tolerance upregulation than other tissues [58]. However, the expression levels of *Mr-grp78* were significantly increased in the following time courses, which may be associated with the functional effects of crustacean hyperglycemic hormone (CHH). This crucial hormone is an important factor for controlling metabolic homeostasis and glucose levels in hemolymph [59].

In many mammals, grp78 has been identified, and Mote et al. (1998) indicated that the upregulation of liver *grp78* mRNA is directly related to higher blood glucose concentrations [60]. A time-course experiment of the CHH protein in the copepod *Tigriopus japonicus* by Kim et al. (2013) revealed that copepod CHH transcript levels are elevated from 6 to 96 h during exposure to 35 °C [61]. Several studies revealed that raising temperature could directly affect alterations in the concentration of minerals and other humoral or cellular components in prawn hemolymph, which included higher lactate, glucose and protein concentrations, increased total hemocytes, severe DNA damage, disturbances in the homeostasis of ions such as Na^+^, K^+^ and Ca^2+^, and an increase in the oxidative stress rate by increasing the activity of ROS [62,63,64,65]. Due to the induction of such effects, the brief upregulation of *Mr-grp78* in hemocytes in early periods may be important to prevent host cellular conditions by protein synthesis regulating and maintaining Ca^2+^ levels to withstand oxidative stress conditions [24,66].

Lower temperature is another physical factor that effectively influences all aquatic animals by decreasing their metabolic rates and respiratory activity. It has been shown that long-term cold-water exposure does not affect the survival rate of prawns but may reduce their growth and feed utilization [67]. Similarly, the results of our study concisely showed low expression or suppression of *Mr-grp78* mRNA in response to cold temperature exposure, which may reduce the levels of Hsps involved in cellular stress functions, particularly in protein synthesis mechanisms [51]. The high upregulation of *Mr-grp78* in response to low temperature may be correlated with negative responses to cellular stress. Qiu et al. (2011) also found that the changes in physiological and immunological activities observed in shrimp hemolymph under low temperature effectively induced decreases in osmolality in the plasma, total protein concentration and total hemocytes. Additionally, low-temperature conditions can effectively cause DNA damage, oxidative stress, changes in osmolality and lipid peroxidation in Pacific white shrimp within a few hours of stimulation [68]. In crustaceans, Hsp27 is generally significantly upregulated at 48–98 hT, and similar results have been found for other Hsp70s that act as counterpart chaperones. Furthermore, in subtropical areas, many aquatic animals have been found to effectively synthesize crucial Hsp molecules, particularly Hsp27 and Hsp70, to maintain a normal state of protein and hormone homeostasis under long-term exposure to cold conditions [54,69].

### 4.5. Functional Analysis of Mr-grp78 under Heat and Cold Temperature Conditions by Gene Knockdown

Recently, RNA interference (RNAi) has been employed to examine the function of grp78, particularly in pathogen–host responses. Functional analyses of grp78 in infectious viral diseases have been the focal points of various research works, particularly in mammals, because it normally functions in the ER and on the cell surface. Furthermore, infections by dengue virus, Japanese encephalitis virus, cytomegalovirus, hepatitis C virus, and West Nile virus have been demonstrated to significantly elevate transcriptional expression levels of Hsp78 or grp78 [70,71,72].

Among crustaceans, the effects of *grp78* silencing on immune defense mechanisms against pathogenic viruses have been clearly demonstrated in Pacific white shrimp and black tiger shrimp [24,25]. These reports showed that *grp78* mRNA expression in this shrimp was successfully silenced in the gills and hepatopancreas and was initially suppressed at 6–48 h. Similar to findings in mammals, knockdown of the *grp78* gene in these two shrimp species resulted in significantly high mortality relative to that found in the control group, and this finding reflects the crucial roles of this gene in the ER and on the cell surface of shrimp cells. Silencing mechanisms also induce severe mortality under conditions in which pH and salinity tolerance is low [25].

In our current study, the transcript levels of *Mr-grp78* were successfully knocked down from 6 or 24 h to 96 h. The *grp78*-silenced prawns were clearly more sensitive to both high temperature (35 °C) and *A. hydrophila* infection than various control prawns. This finding suggests that the absence of Mr-grp78 seriously affects the heat tolerance and immune defense mechanisms of giant river prawns. Further experiments of all consequent mechanisms affected by *Mr-grp78* silencing are needed to clearly explain these mysterious effects. Unexpectedly, ds*GFP* exerted some effects in both the heat- and bacterial-induction experiments, and these findings may be caused by non-target effects of ds*GFP*, which have been observed in some previous reports in invertebrates, such as honeybees [73]. Further information about this phenomenon is needed to clarify this unexpected effect.

## 5. Conclusions

In conclusion, we successfully cloned and characterized the full-length cDNA of the giant river prawn *Mr-grp78* gene. Mr-grp78 has conserved amino acid motifs corresponding to the defined signature of this protein family. Characterization and sequence analyses indicated that the *Mr-grp78* gene exhibits very high similarity scores with sequences of invertebrates, and this finding supports the notion that this gene is highly conserved among organisms. A tissue distribution analysis revealed that the *Mr-grp78* gene was ubiquitously expressed, and high expression was observed in the ovary. Importantly, regulation and silencing analyses based on *Mr-grp78* gene knockdown in the hepatopancreas, gills and hemocytes strongly indicated that this gene plays an important role in cellular stress responses in the ER under bacterial infections and temperature stress.

## Figures and Tables

**Figure 1 animals-11-03004-f001:**
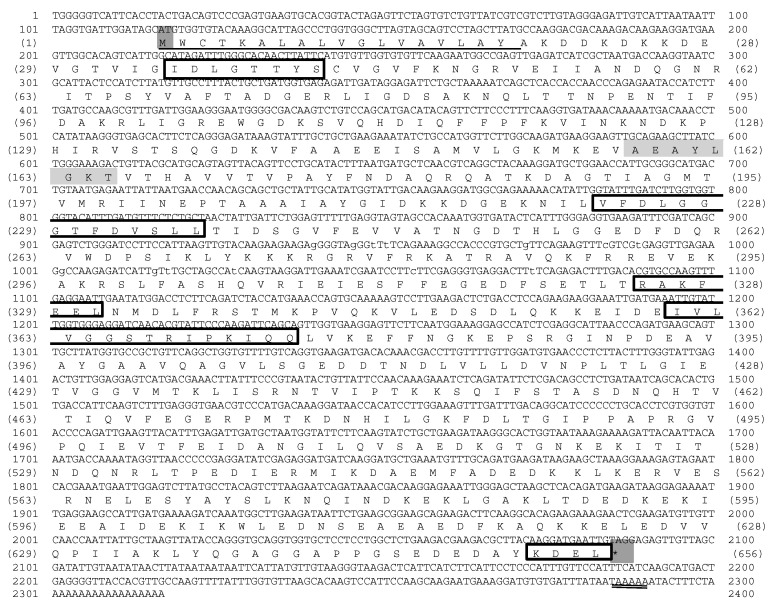
Sequence analyses of the Mr-grp78 gene. The three conserved amino acid signature motifs of the hsp70 family (IDLGTTYS, VFDLGGGTFDVSLL and IVLVGGSTRIPKIQQ), an ER homolog sequence and KDEL (endoplasmic reticulum retention sequence at the C-terminal part) are boxed. A typical signal peptide is underlined, and the *p*-loop (ATP/GTP binding site motif A) is highlighted with a gray background. A polyadenylation signal site (AATAAA) is indicated by double underlining.

**Figure 2 animals-11-03004-f002:**
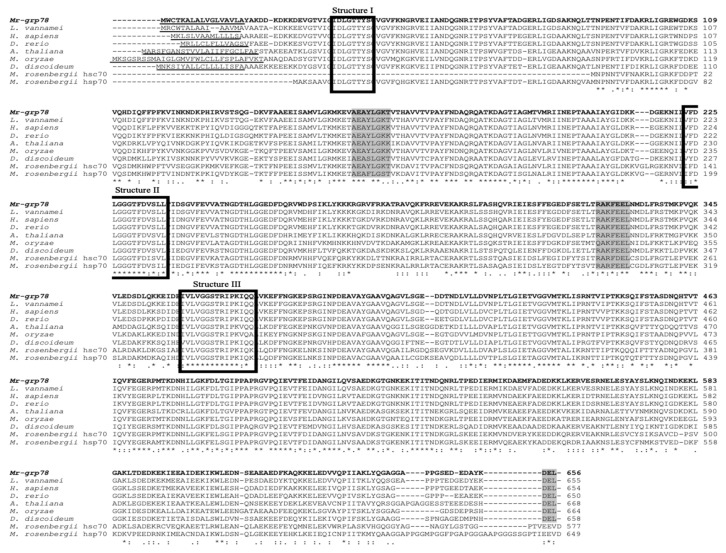
Multiple sequence analyses of Mr-grp78 with grp78s of other known species. The signature sequences of the Hsp70 family and NDEL conserved sequences located at the C-terminal region are boxed. The symbols ., : and * in the alignment represent a semi-conserved substitution, a conserved substitution and an identical residue, respectively. The other Hsp70 sequences of other known species were retrieved from the GenBank database (see Figure 3 for their accession numbers).

**Figure 3 animals-11-03004-f003:**
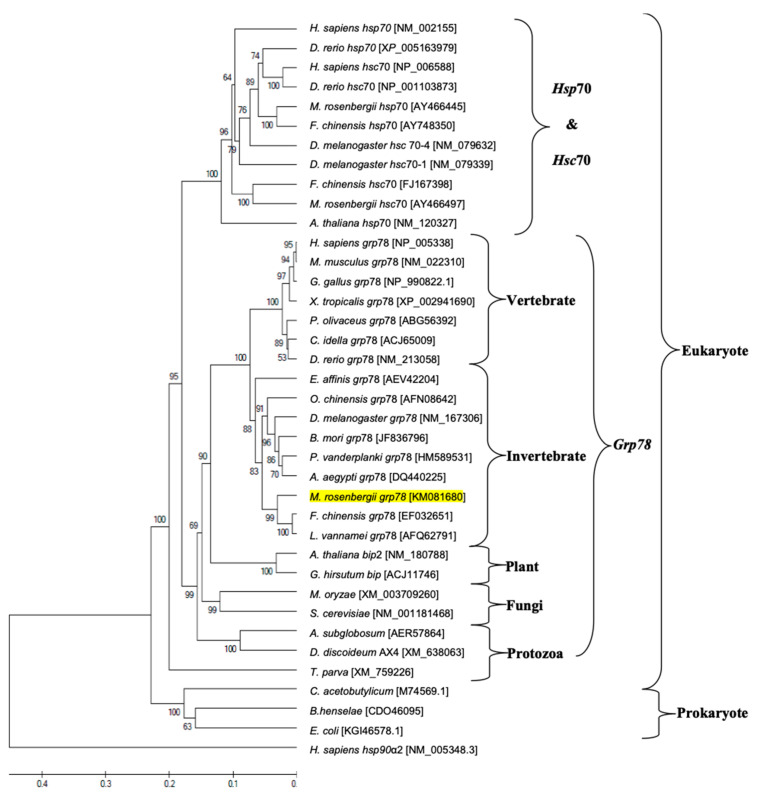
Phylogenetic tree showing the relationship of the *grp78* gene of giant river prawn with that of other known species according to their amino acid sequences available in the GenBank database. GenBank accession numbers of each sequence are indicated next to their species names similar to those detailed in Appendix A. Mr-grp78 is indicated by yellowish shading.

**Figure 4 animals-11-03004-f004:**
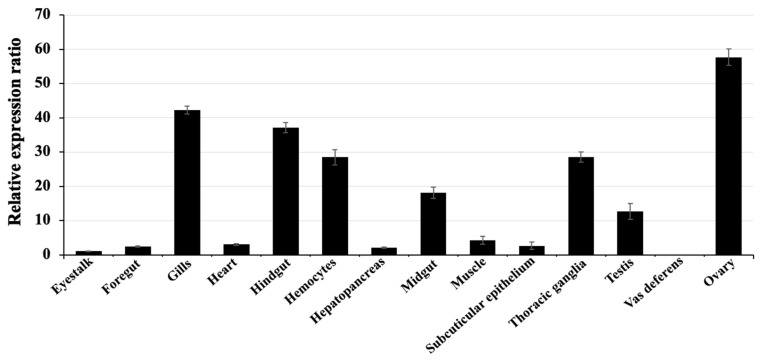
Analysis of *Mr-grp78* expression in 13 tissues by qRT-PCR.

**Figure 5 animals-11-03004-f005:**
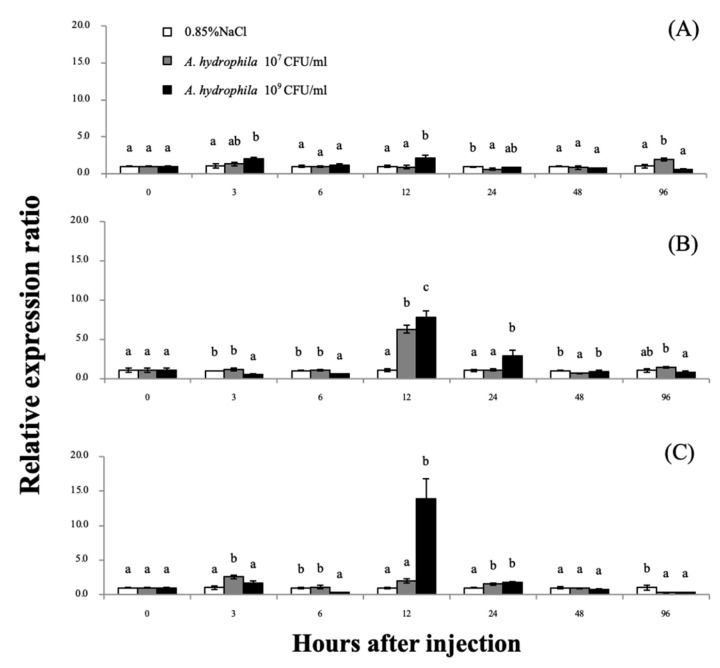
Analysis of *Mr-grp78* mRNA expression in the gills (**A**), hepatopancreas (**B**) and hemocytes (**C**) in response to the injection of *Aeromonas hydrophila* at 2 different concentrations. Different letters among the bars at different time courses indicate significant differences (*p* < 0.05).

**Figure 6 animals-11-03004-f006:**
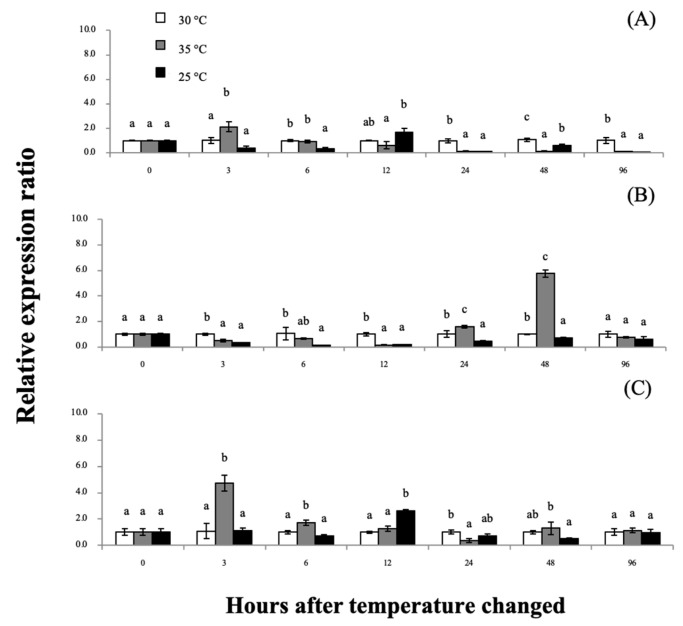
Analysis of *Mr-grp78* mRNA expression in the gills (**A**), hepatopancreas (**B**) and hemocytes (**C**) in response to heat-shock (35 °C) and cold-shock (25 °C) conditions. Different letters among the bars at different time courses indicate significant differences (*p* < 0.05).

**Figure 7 animals-11-03004-f007:**
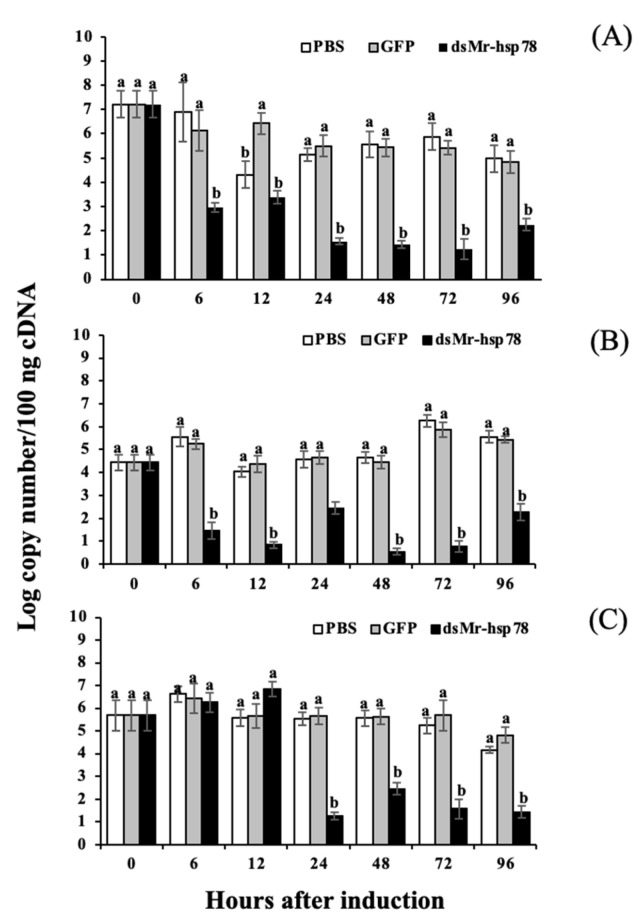
Silencing analysis following gene knockdown of the *Mr-grp78* mRNA transcript in the gills (**A**), hepatopancreas (**B**) and muscle (**C**). Different letters among the bars at different time courses indicate significant differences (*p* < 0.05).

**Figure 8 animals-11-03004-f008:**
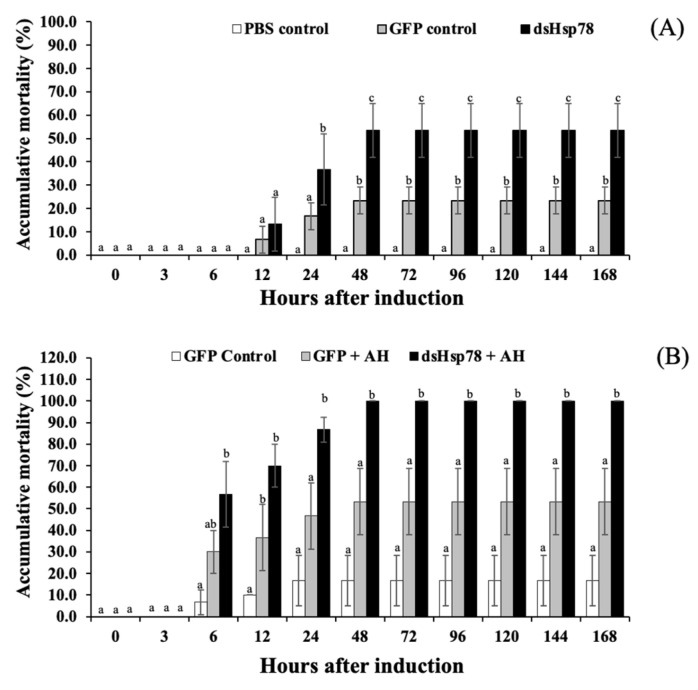
Effects of *Mr-grp78* gene silencing on the mortality of giant freshwater prawns under heat shock (**A**) and *A. hydrophila* infection (**B**). Different letters among the bars at different time courses indicate significant differences (*p* < 0.05).

**Table 1 animals-11-03004-t001:** PCR primers used in the current study.

Gene	Primer Name	Primer Sequence (5’-3’)	Size	Purposes
*β-actin*	RT*B-actin*_F	TTCACCATCGGCATTGAGAGGTTC	119 bp	Real-Time PCR
	RT*B-actin*_R	CACGTCGCACTTCATGATGGAGTT		Real-Time PCR
*Mr-grp78*	F_GFP_*grp78*	TCTGCTGAAGATAAGGGCACTGGT	524 bp	RT-RCR
	R_GFP_*grp78*	ACCTGCACCCTGGTATAACTTAGC		RT-RCR, Touchdown PCR in RACE
	RTF_GFP_*grp78*	TCTGCTGAAGATAAGGGCACTGGT	110 bp	Real-Time PCR
	RTR_GFP_*grp78*	TCAGCATCCTTGATCATCCTCTCG		Real-Time PCR
	Mr-HSP78T7_F	GGATCCTAATACGACTCACTATAGGGGCGACAAGTCTGTCCAGCA	272 bp	Gene silencing
	Mr-HSP78T7_R	GGATCCTAATACGACTCACTATAGGACAGTCATGCCCGCAATGGT		Gene silencing
*Green fluorescence protein* (*GFP*)	GFP_F	TAATACGACTCACTAAGGGAGACACATGAAGCAGCACGACCT		Gene silencing
	GFP_R	TAATACGACTCACTATAGGGAGAAGTTCACCTTGATGCCGTTC		Gene silencing
UPM	LongUPM primer	CTAATACGACTCACTATAGGGCAAGCAGTGGTAACAACGCAGAGT		Touchdown PCR in RACE
	ShortUPM primer	CTAATACGACTCACTATAGGGC		

## Data Availability

Data available on request due to restrictions eg privacy or ethical. The data presented in this study are available on request from the corresponding author. The data are not publicly available due to their containing information that could compromise the privacy of research participants.

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
