# Peer review of "Characterization, Stress Response and Functional Analyses of Giant River Prawn (Macrobrachium rosenbergii) Glucose-Regulated Protein 78 (Mr-grp78) under Temperature Stress and during Aeromonas hydrophila Infection"

_animals, 2021, doi:10.3390/ani11103004_

Round 1

Reviewer 1 Report

Review for the Manuscript animals-1332940

Title: Characterization, Stress Response and Functional Analyses of Glucose-Regulated Protein 78 (Mr-grp78) under Temperature Stress and Aeromonas hydrophila Infection in Giant River Prawn (Macrobrachium rosenbergii, De Man)

General Comments: The manuscript cloned and characterized a glucose-regulated protein 78 (Mr-grp78) encoding gene in the giant river prawn Macrobrachium rosenbergii. Through the use of RT-qPCR, the authors explor its distribution in various tissues and track their response to variations in temperature and pathogenic bacterial exposure to further characterize their biofunctional role. Together with gene knockdown results, the authors suggested that the new yielded Mr-grp78 gene played roles in cellular stress responses to both temperature shock and pathogenic bacterial stimuli. The introduction, the materials and the method sections were generally well written and contained necessary information. Please consider the following comments when the manuscript is revised.

  1. The basis for selecting the targeted gene, glucose-regulated protein 78, in the present study is unclear.
  2. Line 5: delete “De Ma”.
  3. Could authors provide the touchdown PCR protocol (listed in Line 122)?

  1. As indicated in lines 102-108: the experimental animals was prepared in non-axenic conditions. Since Hsp70 is highly conserved protein family, the authors should demonstrate that the new generated Mr-grp78 here is encoded by the DNA of Macrobrachium rosenbergii.

  1. The methods of “Phylogenetic analysis of grp78 genes of various animal species” in Lines 151-157 are insufficient. Please provide more detailed information.

  1. Lines 109-123: The preparation of the template used in 5’-RACE is murky. Please clarify or revise.

  1. Did the authors find any difference in the morphology of the Macrobrachium rosenbergii due to stress (temperature and pathogenic bacterial exposure) ?
  2. The authors used β-actin as a reference gene for the quantitative PCR approach. Actin may not be a good choice for some organisms. It is not described whether the feasibility of actin as a reference gene in Macrobrachium rosenbergii was checked under the chosen conditions.

  1. The Figure 1 and Figure 2 is incomplete in the revision.

10: I think that the expression: " Mr-grp78” in roman type indicates the proteins. If so, the gene symbol should be in italic type.

11: Discussion: I suggest some discussion on this aspect to tie the study back to the environment more coherently. These types of studies are certainly useful for contextualizing genetic response to changing environmental parameters, but I think it is important to also tie back to what the cell really experiences in the wild and how the study informs on the species in this context.

Author Response

Reviewer’s 1

Comments and Suggestions for Authors

Review for the Manuscript animals-1332940

Title: Characterization, Stress Response and Functional Analyses of Glucose-Regulated Protein 78 (Mr-grp78) under Temperature Stress and Aeromonas hydrophila Infection in Giant River Prawn (Macrobrachium rosenbergii, De Man)

General Comments: The manuscript cloned and characterized a glucose-regulated protein 78 (Mr-grp78) encoding gene in the giant river prawn Macrobrachium rosenbergii. Through the use of RT-qPCR, the authors explor its distribution in various tissues and track their response to variations in temperature and pathogenic bacterial exposure to further characterize their biofunctional role. Together with gene knockdown results, the authors suggested that the new yielded Mr-grp78 gene played roles in cellular stress responses to both temperature shock and pathogenic bacterial stimuli. The introduction, the materials and the method sections were generally well written and contained necessary information. Please consider the following comments when the manuscript is revised.

  1. The basis for selecting the targeted gene, glucose-regulated protein 78, in the present study is unclear.

Response: Thank you so much for this suggestion. We have already added up this information in the abstract and last paragraph of introduction section.

  1. Line 5: delete “De Ma”.

Response: “De Man” was completely deleted.

  1. Could authors provide the touchdown PCR protocol (listed in Line 122)?

 Response: We have added this crucial information into section “2.2”.

  1. As indicated in lines 102-108: the experimental animals was prepared in non-axenic conditions. Since Hsp70 is highly conserved protein family, the authors should demonstrate that the new generated Mr-grp78 here is encoded by the DNA of Macrobrachium rosenbergii.

Response: Thank you so much for this comment. We agree with this suggestion of the reviewer, and we always carefully consider this critical point. In our lab, we cloned and characterized various heat shock protein 70s of giant river prawn, and several Mr-hsp70 members have been identified. Therefore, to avoid the miss amplifying of the target gene fragments, all specifical primers have been carefully designed based on the non-conserved nucleotide regions of the target genes.   

  1. The methods of “Phylogenetic analysis of grp78 genes of various animal species” in Lines 151-157 are insufficient. Please provide more detailed information.

Response: Thank you so much for your kind comments. Information in this part has been appropriately modified.

  1. Lines 109-123: The preparation of the template used in 5’-RACE is murky. Please clarify or revise.

Response: This part has been properly modified.

  1. Did the authors find any difference in the morphology of the Macrobrachium rosenbergii due to stress (temperature and pathogenic bacterial exposure)?

Response: Thank you so much for reminding us to put this important information. This suggestion has already been added into sections “3.4” and “3.5”.  

  1. The authors used β-actin as a reference gene for the quantitative PCR approach. Actin may not be a good choice for some organisms. It is not described whether the feasibility of actin as a reference gene in Macrobrachium rosenbergii was checked under the chosen conditions.

Response: Thank you so much for this suggestion. Based on the report by Priyadarshi et al. (2015) and our previous research works. We confirm that β-actin is one of the most suitable candidates housekeeping gene for validation and normalization in RT- or qRT-PCR analysis in the giant river prawn compared to EF1α, GAPDH, and 18S rRNA. The low precision of normalization of β-actin was found when the tissues from the nerve cord were used, and therefore this tissue was not included in the expression responses of our present study.    

  1. The Figure 1 and Figure 2 is incomplete in the revision.

 Response: This part has been properly modified.

10: I think that the expression: " Mr-grp78” in roman type indicates the proteins. If so, the gene symbol should be in italic type.

Response: This part has been properly modified.

11: Discussion: I suggest some discussion on this aspect to tie the study back to the environment more coherently. These types of studies are certainly useful for contextualizing genetic response to changing environmental parameters, but I think it is important to also tie back to what the cell really experiences in the wild and how the study informs on the species in this context.

Response: Thank you for this insightful comment. Some crucial discussions have been properly revised in this part.

Reviewer 2 Report

The authors investigated the stress response and functional analyses of glucose-regulated protein 78 (mr-grp78) under temperature stress and Aeromonas hydrophila infection in giant river prawn. This manuscript (MS) was clearly written and easy to understand. This work can help the sustainability of this species farming. However, some major issues significantly compromised the quality of this MS.

Major comments:

  • First, the manuscript needs to be edited by a native English speaker to improve the language of the MS and fix errors. I mentioned some of them, but still, more works needed to be done.
  • Other aspects such as inappropriate use of abbreviations, absence of scientific names for species and relevant references need to be improved for readers.
  • The authors need to study well enough about some basic biological concepts and pathways such ER, internal stressors, ER stress, misfolding proteins, unfolding proteins, proteolysis etc. Lack of enough knowledge has caused some texts of this MS to be wrongly discussed. Please read the literature more comprehensively and then revise the MS.

However, I have touched on some more points that can contribute to the improvement of this MS.

Minor comments

Abstract

  • Line 17-21, too long; please split it.
  • I suggest changing the topic to present the topic better as you measured many genes plus grp78.
  • Line 23, please revise this part.
  • Line 25, “Biofunctional analysis by grp78 gene knockdown was performed” is not clear; please review it.
  • It is better to start the abstract with a sentence about why did you analyze this work. Then, shortly explain M&M.
  • Line 29-30, please delete this part.
  • Line 29, Please give an introductory text on why Mr-grp78 is important.
  • Line 41, 6-0 hr after what?
  • Line 44, you had a different response in temperature and pathogen treatments. Therefore, you can not sum up as “cellular stress responses”. Throughout the MS, please keep this point in mind.
  • Please reorder the keywords alphabetically and capitalize each word.

Introduction:

  • Well-developed introduction and included a clear fellow and relevant points.
  • Line 51, and misfolding, protein refolding and misfolding are quite different concepts; please check the MS that you use them correctly.
  • Line 52-54, please delete this part.
  • Line 59, stimulated by many stressors, such
  • Line 63, it has many more functions. Please read the literature more comprehensively and complete this part.
  • Line 66, it happens under internal stresses as well; please read more about internal and external stressors and complete this part.
  • Line 78, please delete “excellent”.
  • Line 83 and elsewhere, please be clear with “prawns”. It can be all shrimps and prawns. I suggest using “giant river prawn” in this MS when you want to mention this species.
  • Line 93 and elsewhere, please first mention the common name plus scientific name, and for the rest of the MS, just report the common name.
  • Please review the literature much more carefully and cite more appropriate references.
  • Please check any single reference with reading it instead of only citing it without knowing about their concepts.

Material and methods

  • Well-organized section. Clear fellow and all required details were provided.
  • Line 103, how you knew there were healthy, please provide evidence.
  • Line 108, please add fat, ash and energy contents as well.
  • Line 230, please mention the method of anesthetization somewhere in M&M.

Results

  • Well-written section, all necessary things have been covered.
  • Line 331, please make sure you used hydrophila italic throughout the MS.
  • Line 336, please make it clear in M&M or result what do you mean exactly by “slightly significantly”.
  • Line 337, please make sure you use superscript correctly in the MS.
  • Fig 4, to be clearer, please add the name of tissue instead of abbreviations.
  • Line 385, as you did both cold and hot stress, it is better to be clear in the MS when you say “heat shock”.

Discussion

  • Put the subheading for the discussion section like results. Also, keep a sequence in subheading for investigated factors, in M&M, result, and discussion.

As a general comment: please focus on fish as hips of references and studies are available, and no need to cite other vertebrates.

  • Please make sure the sentences are not too long.
  • Line 446-452, please split this sentence.
  • Line 468, I am not sure for which species is. However, it does not make sense for your species. Please make sure you cited and discussed the relevant studies.
  • Line 468-470, please delete this part.
  • Line 474, scientific name? Please see my comment for the common name and scientific name.
  • Line 503, please make sure you have defined the abbreviations for the first time in the MS.
  • Line 519, please double check this part.
  • Line 522, again, what do you mean by “slight effect”. Please revise the MS from this point and make sure you are so clear about significant differences and changes,
  • Line 559, please delete “in the present study”.
  • Although you wrote this section well, you can still improve it by answering these questions and annotated them to the discussion section. Why were these results observed? Discuss more possible reasons.
  • The conclusion needs to be revised and add more comprehensive concepts there.

Best regards

Author Response

Reviewer’s 2

Comments and Suggestions for Authors

The authors investigated the stress response and functional analyses of glucose-regulated protein 78 (mr-grp78) under temperature stress and Aeromonas hydrophila infection in giant river prawn. This manuscript (MS) was clearly written and easy to understand. This work can help the sustainability of this species farming. However, some major issues significantly compromised the quality of this MS.

Major comments:

First, the manuscript needs to be edited by a native English speaker to improve the language of the MS and fix errors. I mentioned some of them, but still, more works needed to be done.

Response: Thank you so much for this suggestion. The latest version of our manuscript was properly polished by a native speaker, who is an expert in this field.

Other aspects such as inappropriate use of abbreviations, absence of scientific names for species and relevant references need to be improved for readers.

Response: This part has been properly modified.

The authors need to study well enough about some basic biological concepts and pathways such ER, internal stressors, ER stress, misfolding proteins, unfolding proteins, proteolysis etc. Lack of enough knowledge has caused some texts of this MS to be wrongly discussed. Please read the literature more comprehensively and then revise the MS.

Response: Thank you so much for this critical comment and it is very useful for us to improve the quality of our manuscript.

However, I have touched on some more points that can contribute to the improvement of this MS.

Minor comments

Abstract

Line 17-21, too long; please split it.

Response: Thank you so much for this comment. This information has been modified.

I suggest changing the topic to present the topic better as you measured many genes plus grp78.

Response: Thank you so much for this comment. In this present work, we mainly target to grp78 in giant river prawn. We would be happy to change the topic, If the reviewer had other better ideas to improve this part

Line 23, please revise this part.

Response: Thank you so much for this comment. This information has been modified.

Line 25, “Biofunctional analysis by grp78 gene knockdown was performed” is not clear; please review it.

Response: Thank you so much for this comment. This information has been modified.

It is better to start the abstract with a sentence about why did you analyze this work. Then, shortly explain M&M.

Response: Thank you so much for this comment. This information has been modified.

Line 29-30, please delete this part.

Line 29, Please give an introductory text on why Mr-grp78 is important.

Response: Thank you so much for this comment and we have deleted and modified it.

Line 41, 6-96 hr after what?

Response: We just want to point out the critical time of gene-knockdown in this part.

Line 44, you had a different response in temperature and pathogen treatments. Therefore, you cannot sum up as “cellular stress responses”. Throughout the MS, please keep this point in mind.

Response: Thank you so much for this comment. We have carefully corrected this critical point.  

Please reorder the keywords alphabetically and capitalize each word.

Response: Thank you so much for this comment. We have reordered them as the reviewer suggestion.

Introduction:

Well-developed introduction and included a clear fellow and relevant points.

Line 51, and misfolding, protein refolding and misfolding are quite different concepts; please check the MS that you use them correctly.

Response: Thank you so much for this comment. We have carefully checked this important point thought out the manuscript.

Line 52-54, please delete this part.

Response: Thank you so much for this comment. We have carefully deleted this part from the manuscript.

Line 63, it has many more functions. Please read the literature more comprehensively and complete this part.

Response: Thank you so much for this comment. We have carefully added more information in this part.

Line 66, it happens under internal stresses as well; please read more about internal and external stressors and complete this part.

Response: Thank you so much for this comment. We have carefully corrected this information in this part.

Line 78, please delete “excellent”.

Response: Thank you so much for this comment. We have carefully corrected this information in this part.

Line 83 and elsewhere, please be clear with “prawns”. It can be all shrimps and prawns. I suggest using “giant river prawn” in this MS when you want to mention this species.

Response: Thank you so much for this comment. We have carefully checked this important point thought out the manuscript.

Line 93 and elsewhere, please first mention the common name plus scientific name, and for the rest of the MS, just report the common name.

Response: Thank you so much for this comment. We have carefully corrected this information in this part.

Please review the literature much more carefully and cite more appropriate references.

Please check any single reference with reading it instead of only citing it without knowing about their concepts.

Response: Thank you so much for this comment. We have carefully corrected this information in this part.

Material and methods

Well-organized section. Clear fellow and all required details were provided.

Line 103, how you knew there were healthy, please provide evidence.

Response: Thank you so much for this comment. We have carefully corrected this information in this part.

Line 108, please add fat, ash and energy contents as well.

Response: Thank you so much for this comment. We have added this information in this part.

Line 230, please mention the method of anesthetization somewhere in M&M.

Response: Thank you so much for this comment. We have added this information in this part.

Results

Well-written section, all necessary things have been covered.

Line 331, please make sure you used hydrophila italic throughout the MS.

Response: Thank you so much for this comment. We have carefully checked this important point thought out the manuscript.

Line 336, please make it clear in M&M or result what do you mean exactly by “slightly significantly”.

Response: Thank you so much for this comment. We have carefully checked and corrected this point.

Line 337, please make sure you use superscript correctly in the MS.

Response: Thank you so much for this comment. We have carefully checked this important point thought out the manuscript.

Fig 4, to be clearer, please add the name of tissue instead of abbreviations.

Response: Thank you so much for this comment. We have carefully corrected this point.

Line 385, as you did both cold and hot stress, it is better to be clear in the MS when you say “heat shock”.

Response: Thank you so much for this comment. We have carefully checked and corrected this point and we found that this part conducted only for heat-shock induction.

Discussion

Put the subheading for the discussion section like results. Also, keep a sequence in subheading for investigated factors, in M&M, result, and discussion.

Response: Thank you so much for this comment. We have carefully checked and modified this important point in this part.

As a general comment: please focus on fish as hips of references and studies are available, and no need to cite other vertebrates.

Response: Thank you so much for this comment. We have carefully checked and kept focusing on describing information of invertebrates in this part.

Please make sure the sentences are not too long.

Line 446-452, please split this sentence.

Response: Thank you so much for this comment. We have carefully checked and modified this important point in this part.

Line 468, I am not sure for which species is. However, it does not make sense for your species. Please make sure you cited and discussed the relevant studies.

Line 468-470, please delete this part.

Response: Thank you so much for this comment. We have carefully checked and modified this important point in this part.

Line 474, scientific name? Please see my comment for the common name and scientific name.

Response: Thank you so much for this comment. We have carefully checked and modified this important point in this part.

Line 503, please make sure you have defined the abbreviations for the first time in the MS.

Response: Thank you so much for this comment. We have carefully checked and ROS was previously mentioned.

Line 519, please double check this part.

Response: Thank you so much for this comment. We have carefully double checked this part.

Line 522, again, what do you mean by “slight effect”. Please revise the MS from this point and make sure you are so clear about significant differences and changes,

Response: Thank you so much for this comment. We have carefully checked and modified this important point.

Line 559, please delete “in the present study”.

Response: Thank you so much for this comment. We have carefully deleted this this content from this part.

Although you wrote this section well, you can still improve it by answering these questions and annotated them to the discussion section. Why were these results observed? Discuss more possible reasons.

The conclusion needs to be revised and add more comprehensive concepts there.

Best regards

Response: Thank you so much for this crucial comment. Based on these suggestions, we have tried to carefully revised this part to meet more comprehensive concepts.

Round 2

Reviewer 1 Report

I support the MS should be published. However,  English could be improved. 

Author Response

Reviewer’s 1

I support the MS should be published. However, English could be improved.

Response: Thank you so much for this suggestion. The latest version of our manuscript was properly polished and proven by a native speaker, who is an expert in this field.

Reviewer 2 Report

The authors improved the quality of the MS and I suggest authors reading one more time to fix few language errors. Then, it would be ready for the final steps for acceptance.

  • Please make sure you addressed my comment regarding the scientific name and common names. I can still see some errors there.
  • Line 338 and elsewhere, make sure you are consistent with hT or hour. I suggest you using “hour” as it is more common than hT
  • In some comments, you only said “We have carefully checked this important point thought out the manuscript” but you did not address them. I strongly suggest that you check all comments in the last version again and address them. For example, in Fig 4, you use still abbreviations. I will check the MS in the next version to see how the comments have been addressed.

Best regards

Author Response

Reviewer’s 2

The authors improved the quality of the MS and I suggest authors reading one more time to fix few language errors. Then, it would be ready for the final steps for acceptance.

Response: Thank you so much for this suggestion. The latest version of our manuscript was properly polished and proven by a native speaker, who is an expert in this field.

Please make sure you addressed my comment regarding the scientific name and common names. I can still see some errors there.

Response: Thank you so much for this suggestion. These trivial errors have been fixed and shown by green highlight in the manuscript.

Line 338 and elsewhere, make sure you are consistent with hT or hour. I suggest you using “hour” as it is more common than hT

Response: Thank you so much for this suggestion. We have properly checked this suggestion and tried to make it is consistent with “hT” and “hI”, because it specifically used to represent “hour after treatment” and “hour after injection”, respectively (Please see green highlights).  

In some comments, you only said “We have carefully checked this important point thought out the manuscript” but you did not address them. I strongly suggest that you check all comments in the last version again and address them. For example, in Fig 4, you use still abbreviations. I will check the MS in the next version to see how the comments have been addressed.

Response: Thank you so much for this suggestion and do apologize for this error. In the last version, we corrected Figure 4. as recommended by the reviewer.

Round 3

Reviewer 2 Report

Accept